Assessing shoreline exposure and oyster habitat suitability maximizes potential success for sustainable shoreline protection using restored oyster reefs

La Peyre Megan K. 1 mlapey@lsu.edu
Serra Kayla 2
Joyner T. Andrew 2
Humphries Austin 3
1 US Geological Survey, Louisiana Cooperative Fish and Wildlife Research Unit, School of Renewable Natural Resources , Baton Rouge, LA , United States
2 Department of Geosciences, Geospatial Exploration Lab, East Tennessee State University , Johnson City, TN , United States
3 Department of Fisheries, Animal and Veterinary Sciences, College of the Environment and Life Sciences, University of Rhode Island , Kingston, RI , United States
Robinson Laura
Electronic publication date: 2015 Oct 6
Publication date: 2015
Volume: 3
Electronic Location ID: e1317
Received 2015 Jul 31; Accepted 2015 Sep 22
Copyright: © 2015 La Peyre et al.
Copyright year: 2015
Copyright holder: La Peyre et al.
License: This is an open access article distributed under the terms of the Creative Commons Attribution License, which permits unrestricted use, distribution, reproduction and adaptation in any medium and for any purpose provided that it is properly attributed. For attribution, the original author(s), title, publication source (PeerJ) and either DOI or URL of the article must be cited.
License URL: https://creativecommons.org/licenses/by/4.0/

Keywords: Natural breakwaters, Gulf of Mexico, Ecosystem services, Living shoreline, Coastal protection, Wave attenuation, Marsh, Crassostrea virginica

Funding: The Nature Conservancy of Louisiana The Louisiana Department of Wildlife and Fisheries Department of Geosciences at East Tennessee State University Data used was made possible through support from The Nature Conservancy of Louisiana through grants to monitor multiple oyster reef restoration projects (Vermilion Bay, Lake Eloi, Lake Fortuna, Grand Isle), and to complete meta-data analyses from these projects. The Louisiana Department of Wildlife and Fisheries supported the project at Sister Lake, and provides support through the US Geological Survey’s Louisiana Fish and Wildlife Cooperative Research Unit. Additional travel funding was provided by the Penn Virginia-Westmoreland Summer Scholarship from the Department of Geosciences at East Tennessee State University. The funders had no role in study design, data collection and analysis, decision to publish, or preparation of the manuscript.

==============================
Oyster reefs provide valuable ecosystem services that contribute to coastal resilience. Unfortunately, many reefs have been degraded or removed completely, and there are increased efforts to restore oysters in many coastal areas. In particular, much attention has recently been given to the restoration of shellfish reefs along eroding shorelines to reduce erosion. Such fringing reef approaches, however, often lack empirical data to identify locations where reefs are most effective in reducing marsh erosion, or fully take into account habitat suitability. Using monitoring data from 5 separate fringing reef projects across coastal Louisiana, we quantify shoreline exposure (fetch + wind direction + wind speed) and reef impacts on shoreline retreat. Our results indicate that fringing oyster reefs have a higher impact on shoreline retreat at higher exposure shorelines. At higher exposures, fringing reefs reduced marsh edge erosion an average of 1.0 m y−1. Using these data, we identify ranges of shoreline exposure values where oyster reefs are most effective at reducing marsh edge erosion and apply this knowledge to a case study within one Louisiana estuary. In Breton Sound estuary, we calculate shoreline exposure at 500 random points and then overlay a habitat suitability index for oysters. This method and the resulting visualization show areas most likely to support sustainable oyster populations as well as significantly reduce shoreline erosion. Our results demonstrate how site selection criteria, which include shoreline exposure and habitat suitability, are critical to ensuring greater positive impacts and longevity of oyster reef restoration projects.

Introduction

The historic loss of structurally complex, three-dimensional oyster reefs has cascading impacts on nearshore ecosystems and communities. For example, it is estimated that 85% of the filtration capacity of oysters in the United States has been lost in the past century (Zu Ermgassen et al., 2012). This has consequences for nutrient dynamics and eutrophication, potentially contributing to larger hypoxic zones. Furthermore, commercial fisheries that target oyster reef associated species may lose a significant portion of their value as a result of habitat degradation (Peterson, Grabowski & Powers, 2003). The consequences of oyster reef loss on marsh erosion are less clear, but significant resources have been recently invested in using these ecosystem engineers as a tool for coastal protection and adaptation (Rodriquez et al., 2014; Arkema et al., 2013; Borsje et al., 2011).

Hundreds of miles of shorelines are protected throughout the United States in efforts to stabilize and prevent the loss of coastal lands, including marshes (Restore America’s Estuaries, 2015). While there are a multitude of approaches used for shoreline protection, recent focus on the use of natural and self-sustaining systems has promoted the development of fringing oyster reefs within estuarine systems (La Peyre et al., 2014; Scyphers et al., 2011; Stricklin et al., 2009). Oyster reefs are promoted as they may combat marsh erosion by altering water flow patterns and attenuating waves (Borsje et al., 2011), and trapping and stabilizing sediment (Walles et al., 2015a; Walles et al., 2015b; Van Leeuwen et al., 2010; Meyer, Townsend & Thayer, 1997). Importantly, oysters may indirectly affect the propagation of waves by building three-dimensional reefs, and altering coastal bathymetry, a primary control of wave energy (Le Hir et al., 2000). Of particular value is that oyster reefs may provide a long-term sustainable solution as they can be self-sustaining, and can produce a crystallizing cement of calcium carbonate (Harper, 1997), which allows individual oysters to bond together and build biogenic carbonate reefs in estuaries (Rodriquez et al., 2014; Waldbusser, Steenson & Green, 2011; Walles et al., 2015a; Walles et al., 2015b).

Despite these hypothesized benefits of fringing oyster reefs, evidence for their impacts on reducing shoreline erosion remain equivocal based on the ultimate metric of changes in marsh edge retreat (i.e., Ysebaert et al., 2012; Scyphers et al., 2011; Piazza, Banks & La Peyre, 2005). Success of these projects, similar to most biologically based restoration projects, is critically dependent on appropriate site selection (Beseres Pollack et al., 2012; Coen & Luckenbach, 2000). In this case, long-term success likely depends on identifying locations suitable for oyster reef sustainability and understanding how shoreline orientation with respect to fetch and dominant wind speed and direction relate to the potential for oyster reefs to mitigate their erosive energies on adjacent marsh edges.

Assessing shoreline exposure to erosive wave energies remains difficult to determine directly in dynamic and remote shallow-water environments. However, the application of GIS and wave models for estuarine systems has proven to be an alternative and useful means to characterize spatial and temporal attributes for shoreline exposure and may remove considerable uncertainty (e.g., Callaghan et al., 2010; Carniello et al., 2005). One particular approach uses GIS and data from nearby continuous data recorders to calculate an index of shoreline exposure based on fetch and wind speed and direction. Different calculations have been used to create an index of exposure to assess the effects of exposure on edge habitat use by fish assemblages (La Peyre & Birdsong, 2008), to examine effects on vegetation zonation along freshwater lake edges (Keddy, 1982), and more recently to examine exposure of shorelines and edge erosion in one Louisiana estuarine lake system (La Peyre et al., 2014). Determining the relationship between marsh edge erosion in relation to both exposure of shorelines, and the presence or absence of shoreline protection structures could prove useful in identifying priority locations for restoration.

Equally important as identifying shoreline exposures where fringing oyster reefs may be effective in reducing erosion, is selecting areas of suitable habitat for sustainable oyster populations (Soniat et al., 2013; Melancon et al., 1998; Cake, 1983; Gunter, 1952). Locations conducive to high production of shell substrate, through settlement and growth are a necessary requirement for longevity (Powell et al., 2012; Walles et al., 2015a; Walles et al., 2015b). Habitat suitability indices (HSI) were developed for environmental impact assessments initially (Cake, 1983), and more recently used for aquaculture, conservation, and restoration applications (Soniat et al., 2013; Beseres Pollack et al., 2012; Cho et al., 2012). For oysters, salinity is a key variable affecting many aspects of its life including growth, mortality, reproduction, predation, and disease infection levels (Shumway, 1996). In a highly variable estuarine environment with significant freshwater inflow, such as in coastal Louisiana, salinity often dominates models related to oyster sustainability, and includes not just mean salinity levels, but timing and range of salinity during critical parts of the year (Soniat et al., 2013). In considering appropriate locations to invest in fringing oyster reefs for shoreline protection, the use of an oyster HSI is important.

Coastal Louisiana is currently experiencing high rates of land loss and efforts to reduce erosion and restore marsh habitat are currently underway, with more plans for the future (i.e., Louisiana’s Comprehensive Master Plan for a Sustainable Coast; CPRA, 2012). The master plan proposes a combination of actions including sediment diversion, marsh creation, shoreline protection, and hydrological restoration. One of the shoreline protection techniques involves the use of fringing oyster reefs to reduce shoreline erosion, and to provide for other ecosystem services (i.e., habitat provision, water quality). We propose a framework to evaluate potential sites for the creation of fringing oyster reefs as a shoreline protection tool. This framework combines habitat suitability of sites for oysters, with shoreline exposure.

Materials and Methods

Using empirical data of lateral marsh movement collected in the field at five sites in Louisiana where fringing oyster reefs were restored, we relate shoreline exposure to shoreline movement measured. We identify ranges of shoreline exposure values where oyster reefs most effectively reduced marsh edge retreat at these five sites (details in (I) Empirical data). We then apply this knowledge to a case study within one Louisiana estuary, Breton Sound (details in (II) Application for restoration planning). In Breton Sound estuary, we calculate shoreline exposure at 500 random points, and overlay an HSI for oysters to visualize areas most likely to support both sustainable oyster reef populations and significantly reduce lateral marsh retreat.

(I) Empirical data

Shoreline movement data

We used shoreline movement data collected between 2009 and 2014 from five fringing reef restoration projects built along a ∼300 km stretch of Louisiana coast: Caillou Lake (locally known, and hereafter referred to, as Sister Lake), Grand Isle, Lake Eloi, Lake Fortuna, and Vermilion Bay (Fig. 1; Table 1; La Peyre et al., 2014; La Peyre, Schwarting & Miller, 2013a; La Peyre, Schwarting & Miller, 2013b; M La Peyre, 2014, unpublished data). All five projects were located in areas that are primarily open-water, brackish systems with a mean tidal range of 0.2–0.5 m, and have low depth profiles with averages of 1.5 m or less under mean water conditions. All areas have historically supported oyster growth and have highly eroding shorelines (LDWF, 2013; Couvillion et al., 2011). In initial monitoring of sites, all sites successfully recruited oyster populations, although apparent survival and growth rates differed by sites (Casas, La Peyre & La Peyre, 2015; La Peyre et al., 2014; La Peyre, Schwarting & Miller, 2013a; La Peyre, Schwarting & Miller, 2013b). Sites were restored using different material for the reef base, but all were narrow, fringing reefs between 0.5 and 1.0 m in height, located between 10 and 50 m from shoreline, along the 1 m contour line, and conceived for shoreline protection as their primary function (Table 1).

Figure 1 Study area map.

Study area map of oyster reef restoration sites in coastal Louisiana used for analyses.

Table 1 Overview of reef projects used for analyses.

Descriptions of created fringing oyster reefs examined in this study for effectiveness in reducing shoreline edge erosion. Independent sample points either represented single reef segments (Sister Lake, Vermilion) or independent and random reef sample points along the reef, a minimum of 200 m from any other sample points (Grand Isle, Lake Eloi, Lake Fortuna). Total project length represents the length of the individual reef segments while height is the constructed height of the reef and represents the maximum possible elevation above the bottom as reefs settle in the soft sediments. Reefs were placed along the 1 m contour line, or within 50 m of the water edge, whichever was closer.

Area	Parish	Number of independent sample points	Total project length (km)/height (m)	Reef base material	Construction date	Sampling events	
Sister Lake	Terrebonne	9	0.23/0.7	Oyster shell	2009 Mar	9	
Vermilion	Vermilion	6	0.41/0.6	Oysterbreak™ rings	2010 Aug	5	
Grand Isle	Jefferson	3	1.4/0.6	Reefblk™ triangles	2011 Apr	8	
Lake Eloi	St. Bernard	3	2.4/0.6	Reefblk™ triangles	2012 Jan	3	
Lake Fortuna	St. Bernard	3	1.3/0.6	Reefblk™ triangles	2012 Jan	3	

All projects measured shoreline movement using similar methods, and individual observations (with sampling intervals spanning less than 12 months) of shoreline movement were included in the analyses (n = 228). Briefly, shoreline position change was measured using techniques similar to Meyer, Townsend & Thayer (1997) and Piazza, Banks & La Peyre (2005). Five transects were established within each site with permanent base stakes located in the marsh and in the water. For each sample, a tape measure was stretched level between base stakes and read at the shoreline edge along the same compass point each time. Shoreline edge is defined as the farthest waterward extent of the emergent wetland macrophytes. Change in shoreline position was calculated as the difference (cm) between measurements. Positive values indicate accretion, negative values indicate erosion. Shoreline retreat for each location and observation period is reported in m yr−1.

Exposure calculations

For each site, wind direction and speed covering the time of the observations were downloaded from the nearest continuous data recorder (Table 1). To identify intricate shoreline details in the Gulf of Mexico a very high resolution (1:5,000) shoreline dataset was obtained from the Gulf of Mexico Coastal Ocean Observing System (GCOOS: www.gcoos.org). The Gulf of Mexico shoreline was derived from the larger National Oceanic and Atmospheric Administration (NOAA) Office of Coastal Management composite shoreline database (www.shoreline.noaa.gov), which covers all of the continental United States and Hawaii. Using this base map, fetch (distance; m) from each observation point to the nearest shoreline was calculated along 16 “bearing lines” based on equal intervals of 22.5 azimuth degrees for each individual observation point (Fig. 2). If the line did not cross any water, that line was given a measurement of zero (i.e., the line was in the direction of land). This was repeated for each point until all measurements were recorded. Exposure was calculated using wind direction, wind speed (km h−1) and fetch (m) and integrating along all 16 bearing lines following methods reported in La Peyre et al. (2014). Wind direction and speed were taken from continuous data recorders using a daily time step (http://waterdata.usgs.gov/la/nwis/—Grand Isle: 073802516; Vermilion Bay: 97387949; Sister Lake: 07381349; and http://www1.ncdc.noaa.gov/—Lake Eloi and Lake Fortuna: NOAA National Climatic Data Center, USWOOO12968). Exposure was calculated using the equation, as written in La Peyre et al. (2014): EM=∑i=116mean wind velocity22.5i°∗percent frequency22.5i°∗fetch22.5i°.

Exposure values for the entire dataset were then divided into three categories based on quartile analysis of normalized exposure values, where ‘low’ and ‘high’ consisted of the lower and upper 25% of values, respectively, and ‘intermediate’ as the middle 50%.

Figure 2 Example of shoreline exposure measurements.

Shoreline exposure was calculated using fetch distance along 16 bearing lines from each sample point (green). Values on each line represent the fetch (distance to closest shoreline) in each direction. Solid gray represents water area while stippled white represents land areas. Gray dots represent other potential random points for site consideration used in the larger data analysis. The light blue line in the inset provides the cut-out of the area used for Breton Sound estuary (i.e., Fig. 4).

Statistical analyses

We used a repeated measures two-way analysis of variance (ANOVA) to evaluate marsh edge erosion in response to (1) oyster reef restoration (reef, mud), (2) the level of exposure (low, intermediate, high), as well as their interactions. Shoreline retreat was analyzed using a generalized linear mixed model with a normal distribution and identity link function (Proc Glimmix, SAS 9.2). The model assumed a randomized block design with sub-sampling within each block (area). We examined the effects of treatment (reef, mud) and exposure (low, intermediate, high), and included random effects of area, and area by material interactions, accounting for replication with each area through a nested statement. Where there were significant interactions, we used main effects models with linear contrasts to determine formal relationships. A significance value of 0.05 was used for all analyses, and means and standard error are reported.

(II) Application for restoration planning

We selected Breton Sound estuary as our study area to develop a framework to identify potential locations where fringing oyster reefs would be most likely to be sustainable and have a significant impact on shoreline retreat. This framework is based on combining our exposure-retreat relationships quantified above, and the habitat suitability index developed for the eastern oyster (Soniat et al., 2013). Breton Sound estuary was selected because it presents a key oyster producing area for Louisiana, and a spatial oyster HSI output had previously been developed (Soniat et al., 2013).

The GCOOS shoreline dataset was clipped to define the extent of the Breton Sound estuary of interest. The northern extent was defined by the targeted 5 ppt isohaline taken from the original Caernarvon Operational Plan (http://coastal.la.gov/wp-content/uploads/2015/01/D.pdf). The 5 ppt isohaline was used as our northern extent because few oysters exist in areas maintained below 5 ppt. Within the defined study area, a total of 500 points were generated within Breton Sound at random locations along the shoreline composite.

For each point, exposure was calculated as described above and in La Peyre et al. (2014), with the exception that wind direction and speed from the last 10 years (2005–2014) were used as a more comprehensive view of longer-term trends. Wind direction and speed data (km h−1) were taken from the New Orleans International Airport recorder (USWOOO12916). Exposure was calculated at each site on a quarterly time-step (Jan–Mar, Apr–Jun, Jul–Sep, Oct–Dec) for each individual year. This time-step was used because it more closely matches the time-step of the field data used above, and, it better captures the low and high exposure periods experienced in the bay due to annual patterns of storms and fronts (Muller & Stone, 2001). This approach yielded 40 exposure values (4 seasons yr−1 × 10 yr) for each of the 500 random points. Each individual exposure value was classified as low, intermediate, or high based on the three quartiles defined using the empirical field data above. Each site was then classified overall as a low, intermediate, or high exposure site using the following rules: if more than 25% of a point’s observations were classified as high exposure, the site was determined to be high exposure. If more than 25% of a point’s exposure values were classified as low, then the site was classified as low exposure. All other sites not meeting either criterion were classified as intermediate exposure sites. Exposure classification for each point was applied to the spatial dataset. The indices were color-coded to reveal high exposure as green, medium exposure as yellow, and low exposure as red.

The resulting exposure classifications for each point were then overlaid on the habitat suitability index (HSI) maps existing for the area (Soniat et al., 2013). The HSI from Soniat et al. (2013) is based on five variables, and calculated on an annual basis as: HSI = (SI1 ∗ SI2 ∗ SI3 ∗ SI4 ∗ SI5)1/5, where SI1 represents bottom habitat type, SI2 represents mean salinity during the spawning season (May–Sept), SI3  represents minimum monthly salinity for the year, SI4 represents annual mean salinity, SI5 represents percent land. An HSI value of 1.0 represents high quality habitat, while HSI = 0 represents poor quality habitat. Explicit details on this HSI can be found in Soniat et al. (2013) and the Louisiana Coastal Master Plan 2012 (http://coastal.la.gov/a-common-vision/2012-coastal-master-plan/cmp-appendices/; Appendix D13).

These HSI maps represent the results of three different simulated discharge “years” showing gridded HSI values for low, average, and high freshwater discharge years. High river discharge years increase freshwater inflow into the estuary and consequently reduce salinity levels. Low discharge years represent the opposite effect. More details on the modeling related to the discharge years can be found in Meselhe et al. (2013a; Meselhe et al., 2013b), and the Louisiana Coastal Master Plan 2012 (http://coastal.la.gov/a-common-vision/2012-coastal-master-plan/cmp-appendices/; Appendix D1). Exposure index values were layered on each of the HSI models to visualize the spatial variability in levels of freshwater discharge and the shift of salinity conditions for oyster sustainability. HSI exhibits a general west-east gradient with higher salinity levels farther into the Gulf of Mexico and lower levels closer to shore. As discharge increases, salinity levels decrease across this gradient and habitat suitability decreases.

Results

(I) Empirical data

Marsh edge erosion was significantly lower at reef sites as compared to mud sites (F1,218 = 12.4, p = 0.0005), and at higher exposure shoreline sites (F2,218 = 3.72, p = 0.0258). There was no significant interaction, and reef base material was not a significant factor. Shorelines categorized as having intermediate and high exposure indices benefited the most from oyster reef restoration (Fig. 3). At high and intermediate exposure sites, shoreline erosion was reduced by an average of 1.07 m yr−1 where reefs were restored when compared to shorelines without oyster reefs.

Figure 3 Shoreline retreat by treatment and exposure.

Mean (± SE) shoreline movement (m yr−1) at low (n = 57), intermediate (n = 113), and high (n = 57) energy shorelines of oyster reef restoration and control mud-bottom treatments at five fringing oyster reef project sites, over three years of data collection across coastal Louisiana (Fig. 1). Means represent shoreline movement on average, over a 3 month period, standardized to m yr−1. Negative values indicate shoreline retreat.

(II) Application for restoration planning

Of the 500 points randomly selected along shorelines in Breton Sound estuary, 156 were classified as high exposure, 190 as intermediate exposure, and 154 as low exposure (Fig. 4). High exposure values (green) indicate shoreline environments where fringing oyster reefs are most likely to reduce shoreline erosion significantly, based on our coast-wide empirical data. Low exposure sites (red) may also benefit from oyster reefs, but the red–orange–green exposure classifications allude to the potential success and impact of oyster reefs ranging from relatively low (red) to relatively high (green) potential for success. Underlying HSI gradients indicate locations and conditions most likely to support sustainable oyster reefs (dark blue), or least likely to support sustainable oyster reefs (light blue), under conditions of low freshwater discharge and salinity into Breton Sound (Fig. 4A), average freshwater discharge and resulting high salinity (Fig. 4B), and high freshwater discharge and resulting low salinity in Breton Sound (Fig. 4C).

Figure 4 Shoreline suitability maps.

Habitat suitability index (HSI) maps with shoreline exposure index points overlaid to visualize optimal areas for oyster reef restoration that would be expected to significantly reduce marsh edge erosion for a (A) high discharge year, (B) low discharge year, and (C) average discharge year. Areas where oyster reef restoration would be expected to have the greatest impact on shoreline erosion and have sustainable oyster populations through time would be where there are green points (high exposure) overlaid on dark blue (high HSI).

Discussion

Ensuring that restoration dollars are effectively spent requires understanding what factors most influence long-term project success and sustainability. Using data from multiple projects and years in coastal Louisiana, fringing oyster reefs were found to effectively reduce marsh retreat by an average of 1 m yr−1 along moderate and high exposure shorelines. Fringing oyster reefs located along sheltered shorelines with low exposure had less impact on marsh retreat rates, although marsh retreat was also lower in these areas. It is important to note that marsh retreat was only reduced in this region, and not reversed and this may be due to other factors, beyond wave energies which also contribute to marsh loss, such as subsidence, and sea level rise (CPRA, 2012). As such, in this area, the use of fringing oyster reefs likely needs to work in combination with other restoration approaches to be fully effective.

Combining shoreline exposure data with oyster habitat suitability models highlight areas where the use of fringing oyster reefs have the greatest potential to provide long-term sustainable oyster reefs while also reducing shoreline retreat. The high variation in habitat suitability index (HSI) results under different freshwater inflow scenarios highlights the importance of considering trade-offs in decision-making related to different restoration and river management options, and the difficulty of incorporating predicted future scenarios in decision-making. However, using a framework that incorporates both oyster habitat suitability, and exposure data helps to better identify specific sites that are more likely to benefit from fringing reefs over the long term by considering likelihood of reef sustainability in conjunction with wind and fetch data. In this area, dominant winds are from the southeast, except during the winter when northerly winds accompany cold fronts. Shorelines with this exposure, combined with a higher fetch resulted, on average, in higher shoreline reduction behind reefs. Within a highly variable environment, with numerous interacting restoration projects, anthropogenic management of important riverine systems and inputs, and unpredictable climate and severe storm events, explicitly identifying factors affecting restoration project outcomes provides critical information.

A challenge for many restoration activities involving living sessile organisms, and fixed project areas revolves around predicting future conditions. HSIs and other habitat assessment models are common tools to study effects of environmental factors on specific species resulting from on-going, potential and predicted land use changes, watershed alteration, and storms (Soniat & Brody, 1988). However, HSI outputs are entirely dependent on the environmental data fed to the model, and as is the case with our Breton Sound example, can result in widely varying outputs. In the case of Breton Sound estuary, the high range of HSI outputs derives largely from the influence of Mississippi River outflow which is dependent on upstream precipitation, and, to some extent by human management of the river inflow into the estuary. Selecting which scenarios to use for restoration decision making can be tricky. At the same time, by combining HSIs under different large-scale scenarios provides managers with the opportunity to make informed decisions, understanding probabilities, and can be useful for decisions on other projects that might affect HSIs. The HSI can serve as a guide for assessing the potential for long-term oyster reef sustainability, but as the HSI outputs are based on inputs from other models (i.e., salinity from an ecohydrology model, Meselhe et al., 2013a; Meselhe et al., 2013b), these inputs themselves come with possible errors and variation (Habib & Reed, 2013), and outputs are dependent on spatial resolution of the model.

While the habitat requirements of oysters are well-known, several HSIs exist for oysters along the Gulf Coast and all vary slightly. For example, Cake (1983) and Soniat & Brody (1988) incorporate historical site salinities into their HSIs for oysters, and this helps better understand long-term site sustainability. Beseres Pollack et al. (2012) provide a restoration HSI which incorporates salinity, temperature, turbidity, dissolved oxygen and depth. The model used for the work presented here, described in detail in Soniat et al. (2013) was calculated on a 1 yr time step thus providing habitat suitability for a given year only. Furthermore, the Soniat et al. (2013) incorporates a variable which references the current habitat bottom type (percent of hard bottom habitat), thus reducing the HSI value over areas or shorelines that currently have only soft mud-bottom. As a result, in many locations, the use of the HSI may underestimate the suitability of a site for restoration purposes, when restoration involves providing substrate; at the same time, practitioners have found that it can be extremely difficult to restore oysters using any engineering materials in any areas in Louisiana characterized solely by soft silts and clays common to these estuaries.

The effects of exposure as measured here on oyster populations themselves have not been fully considered. A meta-data analysis of the effects of exposure on the five projects used to analyze marsh retreat is not informative due to the variation in salinity and other location-dependent data that are known to influence oyster populations themselves (La Peyre et al., 2014; La Peyre, Schwarting & Miller, 2013a; La Peyre, Schwarting & Miller, 2013b). Research does indicate that water flow rates can influence key life-history processes on oyster reefs including recruitment and growth (through food delivery and feeding rates (Lenihan, Peterson & Allen, 1996; Walne, 1972)). However, if there are ideal flow velocities or a threshold above which oysters may not settle or feed, they remain to be quantified (Dame, 2012; Newell & Langdon, 1996; Grizzle & Lutz, 1989). Better parameterizing these relationships would help improve the model, as would better understanding oyster metapopulation dynamics, particularly when establishing new reef locations.

The interaction of both changes in reef height and position based on the development (or lack of) of an oyster reef is also important to consider. For example, if marsh retreat continues, the distance between the fringing reef and marsh edge may increase, thereby reducing the potential effectiveness of the reef over time. The ability of reefs to accrete over time is thus critical, and can be dependent on factors not captured in the basic HSI model used, including basic demographic rates of the eastern oyster (Mann & Powell, 2007), tidal prisms (Byers et al., 2015) and local subsidence and sedimentation rates (Mann & Powell, 2007). No impacts of the base material used for creation were detected in this study, nor was reef height or distance of reef from edge a variable factor as all reefs were of similar height, and location in relation to marsh edge. In coastal Louisiana, marsh erosion rates are high, and are predicted to continue even with significant restoration activities planned (CPRA, 2012). Unless the reefs can expand and grow in order to maintain not only relative height, but distances from changing marsh edges, their ability to provide positive shoreline protection benefits may be limited.

Other factors, aside from oyster suitability, shoreline exposure, and local change patterns may be important to incorporate into this framework for site selection. For example, Walles et al. (2015a) and Walles et al. (2015b) considered the effects of reefs beyond their actual structure and found that in their study area (the Netherlands), elevated areas of mud bottom of approximately the same footprint of the reef, occurred on the lee side of reefs. They found that this elevated area was most related to reef length, suggesting that longer reefs may be more likely to have positive impacts. Other characteristics, including reef shape, and adjacent habitats may also need to be evaluated for inclusion in a full model.

Restoration of coastal habitats can involve multiple trade-offs, not only within one project, but between multiple restoration and river management projects. To use restoration dollars most effectively inherently requires analysis of past projects, and projections of outcomes under future scenarios. In this case, multiple year and project data were examined to identify where fringing oyster reefs were most effective in reducing marsh retreat; these projects were established using different restoration techniques, and had variable success in building a large oyster population (La Peyre et al., 2014; La Peyre, Schwarting & Miller, 2013a; La Peyre, Schwarting & Miller, 2013b). The combination of on-going ecohydrology modeling, HSI models and project monitoring data provides for a quantitative approach to assist stakeholders and managers in planning for future fringing oyster reef projects, maximizing for long-term sustainability. This approach provides a framework for application in other regions, and may be highly useful to evaluate site-specific suitability for small scale shoreline protection structures (i.e., Scyphers, Powers & Heck, 2015).

Supplemental Information

Supplemental Information 1 Data for site exposure and erosion

Data used for initial analyses. More complete datasets available from La Peyre, Schwarting & Miller (2013a), La Peyre, Schwarting & Miller (2013b) and La Peyre et al. (2014).

Click here for additional data file.

Numerous students, research associates and collaborators worked on the original monitoring projects, providing the large dataset for analysis. Thanks to Lindsay Schwarting Miller for help with dataset compilations. Thanks to comments from Donna Bilkovic, Stephen Scyphers, Brenda Walles and one anonymous reviewer who provided comments that significantly improved this manuscript. Any use of trade, firm, or product names is for descriptive purposes only and does not imply endorsement by the US Government.

Additional Information and Declarations

Competing Interests

Author Contributions

The authors declare there are no competing interests.

Megan K. La Peyre and Austin Humphries conceived and designed the experiments, performed the experiments, analyzed the data, contributed reagents/materials/analysis tools, wrote the paper, prepared figures and/or tables, reviewed drafts of the paper.

Kayla Serra and T. Andrew Joyner performed the experiments, analyzed the data, contributed reagents/materials/analysis tools, wrote the paper, prepared figures and/or tables, reviewed drafts of the paper.

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
