# Peer review of "Assessing shoreline exposure and oyster habitat suitability maximizes potential success for sustainable shoreline protection using restored oyster reefs"

_PeerJ, doi:10.7717/peerj.1317_

## Round 0.1 · original submission · Minor Revisions

The reviewers commended the clear and concise writing style, and all included suggestions for further improving the manuscript. My impressions is that these edits should be readily achievable, and thus requires minor / moderate revisions before publication.

Reviewer 1 ·

Basic reporting

The article is well written and structured.

Experimental design

More attention needs to be placed on presenting information on the shoreline-change analyses. What is the error associated with the shoreline erosion rates? It is not clear to me how shoreline erosion rates were measured. Is this from aerial photos? If so, what is the resolution of those images and digitizing error? Are the rates determined from field measurements? If this is just from previously-published work (I think that’s the case), then some table needs to be included that summarizes how shoreline movement was measured at each site and errors associated with that measure. Shoreline change is a very important part of the study. The time step is only 5 years so it is important to present how much change was recorded and if that change is above the resolution of the data set(s).

Variations in the height of the reefs and distance away from the shoreline are important to note because those factors determine the success of a reef at mitigating shoreline erosion. It would be nice to include some additional information on that.

Validity of the findings

A main finding of this work is that fringing oyster reefs were found to effectively reduce marsh retreat an average of 1 m/yr along moderate and high exposure shoreline and fringing oyster reefs located along sheltered shorelines with low exposure failed to impact marsh retreat rates. I think most people would have thought this to be the case...if you harden two shorelines, one that is rapidly eroding and one that is slowly eroding, you will see the most change along the shoreline that is rapidly eroding. What is key is that these shorelines are restored oyster reefs, so they should be growing and increasing their footprint and elevation through time, which is very important for coastal LA where rates of relative sea-level rise are high. I would have liked to have seen more information on the function of these restored reefs in terms of oyster densities. If they are productive reefs and are reducing shoreline erosion (as opposed to being negatively impacted from the high energy conditions, like piles of shell or cement) then that is a much more significant finding. The authors do state, at the end of the article, that oyster populations need to be considered. Perhaps that is work in progress.

·

Basic reporting

Peyre and colleagues combined shoreline exposure data with oyster habitat suitability to indicate where oyster reefs could add to shoreline protection. No doubt the manuscript could be useful to other academics and pratitioners interested in shoreline restoration project including biogenic habitats.

Overall, I liked this submission and would find it to be a useful citation after a number of clarifications (and perhaps edits) are made. Independently, none of my comments might rank as "major", but together, the issues I noted in the manuscript did hinder me from getting as much out of this submission as I think is possible. I hope the authors will be able to consider and respond to the detailed comments I have provided below, and then have the paper accepted following a moderate revision.

Their using monitoring data from 5 separate fringing reef projects to quantify shoreline exposure and reef impact on shoreline retreat. the title suggest that they also include oyster habitat suitability, which I really like. I think however you missed the opportunity to have some words on the reef within the 5 projects. Are these reef developing and placed at the wright location using your model data? I miss an in-depth discussion evaluating the locations of the 5 separate fringing reef project. This way you could test project outcome with your modelled outcome based on the Breton Sound.

Your results indicate that reefs reduce marsh erosion and are most efficient at higher exposure shorelines, but still erosion is continuing. You do not make any remark on this. I think it is important to say a few words on this. If the distance between the reef and the edge of the marsh becomes larger due to this marsh erosion, the reef might not be able to protect the marsh anymore.

Furthermore, the discussion mainly focus on HSI, please elaborate more on the combination of shoreline exposure and oyster habitat suitability models, see the paper of scyphers for example.

Specific comments:
Abstract
Line 45: the keywords climate adaptation, storm surge, vulnerability are not really mentioned in the manuscript I would suggest to drop it.

Introduction:
Line 69-72: Walles et al. 2015 is a recent citation for the self-sustainability of oyster reefs and formation of biogenic carbonate reefs in estuaries.
Line 81: edge should be edges.
Line 99: The recent paper of Walles et al. 2015 highlight the necessary requirement of high shell production for longevity.
Line 107: I really like this.
Line 114: … and to provide for other ecosystem services… such as? Perhaps place a few important ones for that area between brackets.

Experimental design

Materials and methods:
Line 138: Could you provide more background information how the projects measured shoreline retreat?
Line 132: space to much before …All five projects and …are primarily

Validity of the findings

Results
Line 228: shoreline erosion was by an average of 1.07 m yr… there is something missing in the sentence. Please rephrase.
Table 1: for reefs which need to attenuate waves to stabilize shorelines, more information on reef characteristics would be interesting. Please add besides the length of the fringing reefs, information on reef relief and width to the reef structure.
Figure 2: What are the grey dots representing? What is the blue line in the inset? It is difficult to distinguish land from water.
Figure 3: do all projects have low intermediate and high exposed locations? Perhaps add something of this in table 1 as well.
Figure 4: add location (Breton Sound).

Discussion:
Line 244: this is not correctly written, please rephrase this first sentence.
Line 247: add “by”… marsh retreat by an average
Line 248: I don’t think “failed” is the wright wording here, please rephrase. You do see the reef has an effect. If you look over the exposure gradient, you see that the impact of the reef increases from low exposure sites to high exposure sites. This has to do with wave attenuation. Under mild conditions small waves will pass the reef and as long as the wave orbit does not tough the top of the reef, waves are not effected and the reef will not protect the marsh. However, higher wave have larger wave orbits and the effect of the reef will be larger. In this the height of the reef is of importance as changes in water depth (due to tides) will determine the moments the reef attenuate waves. In this light, more information on the height of the reefs, and wave patterns could give insight in the effects of the reefs in the 5 projects.
Line 251: As the reefs do not stabilize nor prevent the loss of coastal lands (Figure 3) it only slows the erosion down. There should be some words on this in the discussion. The discussion is now too much on the HSI, please elaborate more on the reef effect as well.

Line 249: The combination of HIS and Exposure index as in Figure 4 could be elaborated on more. Can you recommend certain areas for reef restoration and shoreline protection? Perhaps a certain shore orientation (e.g. South) or do you see certain wind dominance causing this exposure? Elaborate more on this.

·

Basic reporting

The manuscript is well written and concise. The introduction provides a nice overview of oyster reef and living shoreline ecosystem services, as well as presents a clear case that our understanding of erosion-related services is lacking.

Experimental design

The combined synthesis of empirical studies and development of a planning tool works well together. The empirical data analyses focus on the protective effect of oyster reefs across a range of exposures, while the planning tool also brings in a habitat suitability index (HSI) for oysters. Both sets of methods are solid and generally clear, but a couple minor clarifications would improve interpretation and reproducibility:
1. Please consider providing the equations for habitat suitability and exposure in the main text.
2. Similarly, including summary values (max, range, etc.,) for the components of exposure (fetch and wind), habitat suitability (salinity, etc.), and river discharge years would be useful.
3. How/why was 500 selected as the number of random points for the exposure calculations?

Validity of the findings

The main empirical finding of the paper is a consistent pattern of reduced shoreline erosion across varying levels of exposure when fringing oyster reefs are restored, and this is well supported by several years of monitoring data across multiple sites.

Although this effect is most pronounced at higher exposures sites, a trend of reduced erosion adjacent reefs is consistent across all exposures, a point that I believe could be highlighted even more clearly in the abstract and discussion. It appears that this trend may not be statistically significant at lower exposures, but even a measurable reduction for a slowly eroding shoreline may still have considerable influence on decision-makers, particularly waterfront residents.

The restoration planning application has clear value, and the authors do a nice job discussing the HSI dimension. The three visualizations of variability in river discharge provide an excellent illustration of the importance of considering environmental variability for restoration planning. However, the discussion is currently heavily focused on oysters and habitat suitability and would benefit from further consideration of the shoreline protection nuances described above.

Minor question/comment:
Was the green (high exposure) to red (low exposure) color scheme selected to reflect potential success? If so, I would reconsider how this might be interpreted since your empirical data suggest low exposure sites may still benefit.

Additional comments

This paper has potential as not only an important contribution to the oyster restoration and living shorelines literature, but also as a valuable tool for planning and implementation. Most of my comments focus on clarifying interpretation of the erosion-related findings and the implications of these outcomes. However, the paper also contributes to a broader conversation on restoration and living shoreline success. The paper already does great job considering how to predict and potentially maximize success from a site-selection perspective, and I agree this is very important for maximizing return on investment. However, I would also encourage the authors to consider the potential implications of their work for scenarios where shoreline management action must be taken at a particular site (i.e., private residences, public parks, etc.). For instance, could an understanding of exposure and oyster habitat suitability at a site help contribute to decisions on reef design, base materials, or implementation?

---

## Round 0.2 · accepted · Accept

Thank you for making the corrections as advised.